# Evaluating Ensemble Learning Methods for Multi-Modal Emotion Recognition Using Sensor Data Fusion

**DOI:** 10.3390/s22155611

**Published:** 2022-07-27

**Authors:** Eman M. G. Younis, Someya Mohsen Zaki, Eiman Kanjo, Essam H. Houssein

**Affiliations:** 1Faculty of Computers and Information Minia University, Minia 61519, Egypt; essam.halim@mu.edu.eg; 2Faculty of Computers and Information Minia University, Al-Obour High Institute for Management, Computers and Information systems, Obour, Cairo 999060, Egypt; someyam@oi.edu.eg or; 3Computing and Technology, Nottingham Trent University (NTU), Nottingham NG1 4FQ, UK; eiman.kanjo@ntu.ac.uk

**Keywords:** ensemble learning, emotion recognition, physiological and environmental, subject independent predictive models for emotion, multi-modal emotion recognition

## Abstract

Automatic recognition of human emotions is not a trivial process. There are many factors affecting emotions internally and externally. Expressing emotions could also be performed in many ways such as text, speech, body gestures or even physiologically by physiological body responses. Emotion detection enables many applications such as adaptive user interfaces, interactive games, and human robot interaction and many more. The availability of advanced technologies such as mobiles, sensors, and data analytics tools led to the ability to collect data from various sources, which enabled researchers to predict human emotions accurately. Most current research uses them in the lab experiments for data collection. In this work, we use direct and real time sensor data to construct a subject-independent (generic) multi-modal emotion prediction model. This research integrates both on-body physiological markers, surrounding sensory data, and emotion measurements to achieve the following goals: (1) Collecting a multi-modal data set including environmental, body responses, and emotions. (2) Creating subject-independent Predictive models of emotional states based on fusing environmental and physiological variables. (3) Assessing ensemble learning methods and comparing their performance for creating a generic subject-independent model for emotion recognition with high accuracy and comparing the results with previous similar research. To achieve that, we conducted a real-world study “in the wild” with physiological and mobile sensors. Collecting the data-set is coming from participants walking around Minia university campus to create accurate predictive models. Various ensemble learning models (Bagging, Boosting, and Stacking) have been used, combining the following base algorithms (K Nearest Neighbor KNN, Decision Tree DT, Random Forest RF, and Support Vector Machine SVM) as base learners and DT as a meta-classifier. The results showed that, the ensemble stacking learner technique gave the best accuracy of 98.2% compared with other variants of ensemble learning methods. On the contrary, bagging and boosting methods gave (96.4%) and (96.6%) accuracy levels respectively.

## 1. Introduction

Emotion is one of the most visual and significant elements of our lives, yet it is also one of the most complex subjects to explain scientifically. The study of emotions has many scientific debates in numerous research fields. Emotion is an expression of the cognitive state of the human mind. The emotional response of the human mind presents itself in many ways, such as facial expression [1], voice [2], gesture, posture, and bio-potential signals or physiological reactions [3].

People may disguise their emotions through facial expressions, speech, and physical gestures, but it is difficult to conceal physiological signs like heart rate (HR). Human-Computer Interaction (HCI), adaptive user interfaces, and many more disciplines rely on emotion recognition. HCI develops more intelligent apps that can respond to user demands automatically based on their emotional states [4]. The capabilities and accessibility of less expensive, more delicate, less intruding, and modern sensors for gases, water quality, noise, and other environmental measurements have increased, enabling researchers to collect data in unprecedented geographical, temporal, and contextual details. Recently, there are many studies concerned with the integration of environmental data models and sensors with human health. Sensors are essential to assess the influence of human activities on the environment and human health. They are used as tools for understanding processes, identifying connections, and establishing correlations between environmental and physiological variables [5].

In this work, we employ a sensor-data-fusion approach to consider the following questions:How can we combine physiological body data with environmental data for emotion recognition?Is it possible to create subject independent emotion prediction model with high accuracy?What is the best algorithm with ensemble learning to use for integrating multi-modal data to generate subject-independent models using sensor data fusion?

The fundamental reason for the ensemble learning comes from human nature to collect and weigh multiple viewpoints to make a complex decision. The primary premise is that evaluating and aggregating multiple individual opinions will be better than choosing only one view [6].

Ensemble approaches depend on the idea of combining numerous basic models to produce a powerful learner (or ensemble model) that delivers better results. For many machine learning problems, ensemble methods are regarded as the state-of-the-art solution. By training many models and integrating their predictions, such strategies increase the predictive performance of a single model. The primary concept of ensemble learning is that by merging many models, the faults of a single inducer will most likely be compensated by other inducers, resulting in the ensemble’s total prediction performance being better than a single inducer [6].

Ensemble approaches often increase prediction performance for various reasons (Dietterich, 2002; Polikar, 2006):**Avoid over-fitting**: When only a small quantity of data is available, a learning algorithm is prone to discovering several diverse hypotheses that perfectly predict all of the training data while producing poor predictions for unknown instances. Averaging different hypothesis reduces the risk of choosing an incorrect hypothesis and also improves the overall predictive performance.**Provide a computation advantage**: Local searches conducted by single learners may become stuck in local optima. Ensemble approaches reduce the risk of attaining a local minimum by mixing numerous learners.

In this work, we collected data from thirty people ’in the wild’ including on-body and environmental variables. The data collected comprising on-body data such as body movement, heart rate (HR), electro-dermal activity (EDA), and body temperature, as well as environmental data such as noise level (Env-noise), air pressure, and ambient light levels (UV) and movement. Furthermore, we obtained user emotion labels via self-report input using the 5-step SAM Scale for Valence. We obtained user emotion labels via self-report input using the 5-step SAM Scale for Valence. GPS data were also acquired during data collection recording user’s position. During data collection. The collected data were obtained, cleaned, aggregated, and smoothed. The Self-Evaluation Manikin (SAM) is a nonverbal visual evaluation instrument that assesses a person’s pleasure, arousal, and dominance in reaction to various stimuli [7].

The rest of the article is organized as follows. Section 2 covers previous work related to emotion detection to quantify the relationship between ambient variables and physiological parameters, as well as a brief discussion of on-Body sensors and related information fusion techniques. The methodology is discussed in Section 3 including data collection, data pre-processing, ensemble predictive models and implementation techniques to create user-independent emotion recognition model. Section 4 illustrates the obtained results of emotion detection models and compare their performance. Section 4.4 discusses Hyper-parameter optimization and provide the optimal parameters of base classifiers for avoiding over-fitting and under-fitting. Section 5 discusses the findings of the work of and Section 6 presents conclusions and future work.

## 2. Related Work

### 2.1. Discussions about Emotion Recognition Using On-Body and Environmental Factors

Smartphones and numerous wearable gadgets, such as smartwatches and wristbands, have been outfitted with sensors capable of continuously monitoring human physiological signals (e.g., heart rate, electrodermal activity, body temperature data) and, in some cases, ambient environment data in recent years (e.g., noise, UV, air pressure, etc.) [8].

Sensors range from customized devices designed for particular purposes those used for more typical individual gadgets, such as cell phones. Individuals may serve as environmental sensors in some situations by publicizing what they see, hear, and feel and participating in public support for environmental factors [9].

In [10] researchers studied the Influence of Weather on affective experience (the link between negative and good emotions and environmental changes in weather such as (temperature, relative humidity, barometric pressure, and luminance).

Park, N. K., & Farr, C. A. (2007) investigated the link between lighting and emotions in a retail setting [11]. Similarly, numerous research projects have investigated emotions and their link with wellbeing and physiological changes; however, only one [12] of them have considered integrating physiological and wellbeing sensors alongside ecological sensors in order to predict and model emotion.

In the past, using wearable sensors in real-world studies was inconvenient and intrusive. Despite this, with the advent of wearable sensors and portable technologies, these sensors have become non-intrusive and acceptable for consumers [13].

Furthermore, many wristbands and wearable devices have sensors that aren’t limited to health or body statistics. Contamination sensors, for example, are readily accessible in a variety of sizes and designs, as are temperature stations and other ecological sensors like light and color sensors [14,15,16]. Examine the Table 1, which contains a list of health sensors used in emotion identification.

These sensors could be combined together and used to predict emotion labels. This is called “**data fusion**” that is described as “the act of combining numerous data sources to generate more consistent, accurate, and valuable information than any data source can give” [17,18].

**Table 1 sensors-22-05611-t001:** List of some on-body sensors that have been used for emotion detection.

Sensor	Signals and Features
Motion	Because modern accelerometers incorporate tri-axial micro-electro-mechanical systems (MEMS) to record three-dimensional acceleration, the motion equation is as follows: $\sqrt{x^{2}+y^{2}+z^{2}}$, where this equation is the root mean square of all three components. In recent years, authors used the accelerometer to identify emotions [19].
Body Temperature	Despite its simplicity, we can use body temperature to gauge a person’s emotions and mood shifts [18,19,20]. Wan-Young Chung demonstrated that variations in skin temperature, known as Temperature Variability (TV), may be used to identify nervous system activity [21].
Heart Rate	The RR interval refers to the period between 2 successive pulse peaks, and the signal produced by this sensor consists of heartbeats. According to many researches, authors sometimes use HR to measure happiness and emotions [20,22,23].
EDA	It is sometimes called Galvanic Skin Resistance (GSR) and is associated with emotional and stress sensitivity [20,24,25,26].

Also, classifying data fusion procedures into three categories (low, middle, and high) depends on the processing step at which fusion occurs.

Low-level data fusion, also known as “data level” fusion, seeks to gather various data components from multiple sensors to complement one another. It is possible to combine external data sources such as user self-reported emotions during data gathering [13,23].During data analysis, intermediate-level data fusion, also known as “Feature Level” fusion, is used to determine the optimal collection of features for classification. For example, the best combination of features, such as EMG, Respiration, Skin Conductance, and ECG, has been retrieved using feature-level fusion [18].Finally, high-level data fusion, often known as “Decision Level” fusion, seeks to enhance decision-making by combining the outcomes of many approaches. Ensemble learning can be considered a decision level fusion.

**Raffaele Gravina** has published research about a study of different data fusion approaches and applications in body sensor networks in [13]. Adrián Colomer Granero* [23] utilized feature level fusion to identify emotions and discovered that the ECG and EDA signals are the most important in emotion categorization.

This study [8] used the data-fusion technique to predict emotion labels and discussed a real-world study using smartphones and wearable devices such that authors used a deep learning approach for emotion classification through an iterative process of adding and eliminating a large number of sensor signals from diverse modalities. It merged the local interactions of three sensor modalities: on-body, environmental, and location, into a global model that reflects signal dynamics plus the temporal linkages between each modality. On the raw sensor data, this method used different learning algorithms, including a hybrid approach that combined Convolutional Neural Network and Long Short-term Memory Recurrent Neural Network (CNN-LSTM). When utilizing a massive number of sensors, the results showed that deep-learning approaches were effective in human emotion classification (average accuracy 95% and F-Measure = 95%), and hybrid models outperformed traditional fully connected deep neural networks (average accuracy 73% and F-Measure = 73%). The hybrid models also beat previously developed Ensemble methods that use feature engineering to train the model (average accuracy 83% and F-Measure = 82%).

And this study [12] developed a user-independent emotion model based on fusing or integrating on-body and environmental sensors using ML algorithms such as: SVM, KNN, RF, DT and aggregated the results of these classifiers using stacking ensemble method.

Based on this, numerous research have now concentrated on creating multi-modal emotion recognition models [12,18,27]. See [28], a recent review about multi-modal emotion detection. Others, looked at creating generic or subject-independent models using different approaches.

### 2.2. Discussions about Emotion Recognition Using Physiological Signals and Facial Expressions

Previously, scientific studies concentrated on recognizing and evaluating emotions by analyzing facial expressions and physiological signals. Some studies used EEG signals for developing generic emotion detection models using traditional ML algorithms (SVM, DT, ANN, and many more) such as [29,30]. In addition, physiological signals have been used for creating subject-independent emotion recognition models in [31,32]. In addition, others used facial expressions to detect emotion labels, whereas others used a combination of facial expressions and physiological signals.

Table 2 depicts an overview of the most widely utilized sensor analysis and feature extraction methods. As a result, the speed of evaluating emotions is highly dependent not only on the measuring method and sensor utilized but also on the data processing and analysis methodology used. Researchers employed different measuring methods and feature extraction techniques based on the next table to evaluate various emotions.

From Table 3, we can deduce that all physiological signals and facial expressions signals were acquired from the wearable sensor to measure physiological signals or facial expressions or wearable smartwatch to measure HR signals, or wearable device to measure all these signals together such as Empatica E4 smartwatch or wristband 2. Finally, we presented an overview about previous works that discuss researches associated with predicting emotions either using environmental and physiological factors or using Facial-expression and physiological sensors.

## 3. Methodology

### 3.1. System Architecture

In this subsection, we produce our framework used in our paper. In this work, we adopted information fusion techniques at three key levels. The proposed architecture is composed of a number of processing steps as shown in Figure 1. First, we identify the problem to solve, emotion recognition. Second, the relevant data is collected using on-body sensors from the ‘Microsoft wrist band and user data from the smartphone (e.g., location, ambient noise, and self-reported emotional states) and then data are combined using data-level fusion. The data is then cleansed and pre-processed in the third step. Fourth, the cleaned data is analyzed using descriptive statistical analysis (mean, median, skewness, kurtosis, and so on), covariance, correlation matrices, and Poincare plots to measure heart rate functions using feature level fusion.

Fifth, to improve the performance of our model, we extract features from pre-processed data sensors using decision-level fusion to extract features and select the most informative attributes and a compact collection of characteristics (feature selection, which eliminates irrelevant features from our model). Then, we split the data-set into two parts: training and testing. Finally, stacking model training and testing are performed to obtain predictive model.

Sixth, the predictive model is evaluated and tested by evaluation measures. Finally, optimization of hyper parameters is used to improve the results until satisfactory performance is obtained.

### 3.2. Data Collection

Participants’ data were collected using a wearable sensor known as the “Microsoft wristband 2” and smartphone app software known as ‘EnvBodySens’ which is wirelessly connected to the Microsoft wristband 2. Data were recorded and stamped with the time and date. The smart band featuring capabilities includes multiple sensors:Monitor your heart rate using an optical heart rate monitor.Accelerometer with three axes.Sensors for Galvanic Skin Response (GPS).Galvanic skin response sensors(EDA).UV sensor.Skin temperature sensor.

Table 4 depicts the data recorded inside the mobile application.

To organize the persistent labeling procedure, we used the 5-step SAM Scale for valence derived from [47]. Posters on the walls and word of mouth urged students in the faculty of Computers and Information to participate in the study. Thirty participants who are all between the ages of 18 and 22 took part in this study. Minia University Ethical Committee gave their ethical approval to the study with code [MU-FCI-22-1].

Participants recorded their data while going around the Minia university campus. Before the data collection session, we provided participants with instructions and data about the study methodology and how to use and wear the wristband correctly.

Participants were then requested to perform a self-report momentary assessment using one of the emotional classes (represented by buttons on the app interface) while walking around the campus with the mobile screen on during the data collection (the screen auto-sleep feature was disabled beforehand). The data were gathered over several days.

### 3.3. Data Pre-Processing

Since this data were collected in a real-time and in-situation setting, there is a need for data cleaning to remove the following types of data errors.

incompleteincorrectirrelevant dataoutliers outliers in data can be identified using a boxplot or a histogram. We define the first quartile of the data as $Q_{1}=X\left[n/4\right]$ and the third quartile of the data as $Q_{3}=X\left[3n/4\right]$. We can compute IQR (Interquartile Range). We also remove columns With Low Variance to eliminate columns with few unique values by filtering columns to be eliminated from the dataset using variance statistics or the variance threshold of each column using specific threshold values ranging from (0.0 to 0.5). Based on these methods, features were reduced from 22 to 18 features. Table 5 represents the extracted and removed features. Figure 2 depicts the connection between the threshold (the x-axis) and the number of filtered features (the y-axis) using Variance Threshold in the modified data-set. Then, we spilled the cleaned data into training and testing sets, with the features scaled using **normalization** or **standardization**.

### 3.4. Ensemble Learning Methods

Since **our research** is investigating the integration of ecological and on-body characteristics using sensor data fusion and ensemble methods. Various ensemble learning methods have been implemented as a decision level fusion method and compared their performance.

So, it was necessary to explain why Ensemble learning methods were used. These approaches were used to improve model predictability by integrating numerous models into a single, highly dependable model. Ensemble approaches reduce bias and variance while increasing model accuracy. In most ensemble systems, base learning is performed with a single algorithm, producing homogeneity among all base learners. Homogeneous base learners share similar properties and are of the same type. In other methods, heterogeneous base learners are used, resulting in heterogeneous ensembles. Different sorts of learners make up heterogeneous base learners [6]. Ensemble learning with meta learners has not been well investigated in the literature, especially for sensor data fusion and emotion recognition. The most popular ensemble methods are boosting, bagging, and stacking as shown in Figure 3.

Single classifiers can sometimes give undesirable results. Furthermore, the amount of data we may evaluate is just too large and complex for a single classifier to handle, as in this study. We employed ensemble models to reduce variance for the bagging (Bootstrap Aggregating) technique, reduce bias for the boosting method, and improve predictive performance of the stacking strategy.

These are machine learning paradigms in which several models (commonly referred to as “weak learners”) are trained to address the same problem and then combined to get better results.

#### 3.4.1. Bagging (Bootstrap Aggregation)

The concept behind bagging is straightforward. They fit several separate models and “average” their predictions to produce a model with lower variance. In practice, they can’t utilize independent models since they need too much data. To fit almost separate models, they rely on the good “approximate qualities” of bootstrap samples (representatively and independently) [6,48]. First, they produce several bootstrap samples such that each of them acts as a separate (nearly) independent data set chosen from the genuine distribution. Then, for each of these samples, they fit a weak learner and aggregate them such that they can “average” their outputs and create an ensemble model with fewer variance than its components. The learned base models are roughly independent and identically distributed (i.i.d.), as the bootstrap samples are also roughly independent and identically distributed (i.i.d.). Finally, by “averaging” the weak learners’ outputs, the predicted answer is not changed, but the variance is reduced.

In other words, Bagging involves fitting several base models to diverse bootstrap samples and constructing an ensemble model that averages the outputs of these weak learners, as shown in Figure 4.

#### 3.4.2. Boosting

Because the several combined weak models are no longer fitted independently from the other in sequential techniques. So, the goal is to train models iteratively such that the training of a specific model is dependent on the models trained at previous steps. The most well-known of these methods is “boosting”, which results in an ensemble model that is less skewed than the weak learners that make it up.

Boosting techniques, like bagging techniques, create a family of models that are then blended to create a learner who is more capable and performs better. When fitting each model in the series, the observations in the dataset that the prior models did a poor job of handling are given greater weight. Boosting is a strategy for placing numerous weak students in a very flexible sequence. Each successive model concentrates its efforts on the most challenging data to fit, resulting in a powerful learner with less bias in the end (even if we can notice that boosting can also have the effect of reducing variance).

Finally, Boosting entails fitting a weak learner continuously, aggregating it to the ensemble model, and updating the training data-set to better account for the current ensemble model’s strengths and shortcomings when fitting the next base model, as shown in Figure 5 [49].

Weak learners can be successively fitted and aggregated once they’ve been chosen depending on two essential boosting algorithms: **adaptive boosting** and **gradient boosting**.

During the sequential process, these two meta-algorithms differ in how they construct and aggregate the weak learners. **Ada-boost** or **Adaptive Boosting** updates the weights linked to each of the training data-set observations at each iteration. In comparison to weights of incorrectly categorised observations, weights of well-classified samples decline. The final ensemble model gives more weight to the models that perform better. As seen in Figure 6 [51], the most popular Ada-boost algorithms are **random forest classifiers** and **decision tree classifiers**, whereas **gradient boosting** changes the value of these observations at each iteration. As shown in Figure 7, weak learners are trained to fit the pseudo-residuals that show which direction to adjust the present ensemble model predictions in order to lower the error. Gradient boosting progressively increases the ensemble’s predictors, allowing earlier forecasters to correct later ones, improving the model’s accuracy. To offset the consequences of errors in the earlier models, new predictors are fitted. The gradient booster can discover and address issues with learners’ predictions thanks to the gradient descent. Decision trees with boosted gradients are used in **xgboost**, which offers faster performance.

#### 3.4.3. Stacking

Stacking is an ensemble learning strategy that uses a meta-classifier to merge numerous classification models. Individual classification models are trained using the entire training set, and the meta-classifier is then fitted using the outputs (meta-features) of the individual classification models in the ensemble. The meta-classifier can be trained using either the predicted class labels or the ensemble probabilities. Stacking is a term used to describe another ensemble method known as a stacked generalization, as shown in Figure 8.

Stacking has been successfully employed in regression, density estimations, distance learning, and classifications.

Stacking differs from bagging and boosting primarily in two ways: first, it frequently examines heterogeneous weak learners (various learning algorithms are merged), whereas bagging and boosting primarily consider homogeneous weak learners. Second, stacking uses a meta-model to combine the underlying models, whereas bagging and boosting use deterministic methods to combine weak learners. Finally, we can compare method of these three ensemble methods as in Figure 9 [52]. Table 6 depicts a summarizing of the difference and characteristics for each ensemble learning method.

### 3.5. Implementation

#### 3.5.1. Feature Extraction

After cleaning, prepossessing, and analyzing data statistically, We used descriptive statistics to derive physiological characteristics For:HR, EDA, bTemp representing in (mean, median, max, min, std and quartiles) [24].For HR, we used a Poincare plot to extract distributions of HRV features and was used to test normality defined in time series and frequency domains [18].We computed the SD1 parameter based on time series as follows: $SD1=\sqrt{\frac{1}{2}SDSD^{2}}$, where SD1 is the standard deviation along the minor axis. On the other hand, SDSD is the standard deviation of successive differences (Time Domain Parameter). Additionally, the SD2 parameter may be calculated as follows: $SD2=\sqrt{2SDNN^{2}-\frac{1}{2}SDSD^{2}}$, where SD2 is the standard deviation on the major axis. And SDNN is the standard deviation of the NNI series(). Thus, a higher heart rate of HRV or a lower heart rate of HRV depends on (SD1/SD2) [53]We also derive frequency domain that indicates the power spectrum of order 12 by integration of low frequency (LF) heartbeats (0.04 to 0.15 Hz) and high-frequency (HF) (0.15 Hz to 0.4 Hz) [54].For Motion: We combined the X, Y, and Z characteristics into a single component called Motion.
(1)$$Motion=\sqrt{X^{2}+Y^{2}+Z^{2}}$$

#### 3.5.2. Feature Fusion Level

We concatenated feature sets from several modalities in this technique to produce two spaces that reflect (the environmental and on-Body modalities). As shown in the data-preprocessing section and the previous section, We extracted 18 features from our data-set. However, not all of these features are important to emotional responses, such that there are features correlated with each other. So, we removed some of them to simplify the model via reducing correlated features. Finally, the model will include the most significant features necessary to explain the emotional response.

#### 3.5.3. Feature Selection

We develop a prediction model in that section to see if we can accurately anticipate a user’s affect state based on contextual and physiological parameters.

We decided to decrease the dimension of characteristics by selecting the most efficient features from 18 features from total of 22 features to make the emotion identification process more effective. We chose the SelectKBest feature selection method, which returned the top k features under an evaluation parameter setting of ‘mutual-info-classif’ or changing the ‘score-func’ parameter (classification problem). SelectKBest was a library function of the sklearn machine learning library implemented in Python.

Furthermore, based on the results of the ‘SelectKBest’ feature selection, we decided to create a predictive model with 12 features that have a strong relationship with the label. Figure 10a depicts the most significant features retrieved from the SelectKBest feature-selection technique. Figure 10b shows the importance of selected features, and we can see that the cumulative significance grows until n = 12 (where n is no. of Features).

#### 3.5.4. Building and Optimizing Classification Models

In this research, we chose the KNN classifier, DT, RF, and SVM as weak learners and decided to use DT as a meta-model. The Decision Tree classifier took as inputs the outputs of the four weak learners and will return the final predictions based on it.

To fit a stacking ensemble composed of L weak learners. We have to follow the steps:Separate the training data into three groups.select L weak learners and fit them to first-fold dataevaluate each of the weak learners for second-foldmake predictions for third-fold observations for each of the L weak learnersfit the meta-model on the third fold, using the weak learners’ predictions as inputs.

Finally, Stacking consists of training a meta-model to produce outputs based on the outputs returned by lower-layer weak learners as shown in Figure 8 [54].

If the model has several hyper-parameters, we must look for the best combination of hyper-parameter values in a multi-dimensional space. That’s why hyper-parameter tuning, or determining the appropriate hyper-parameter values, is such a difficult and time-consuming task. K-fold cross-validation is the most popular type of cross-validation. It’s an iterative method for dividing train data into k divisions. One division is saved for testing in each iteration, while the remaining k-divisions are used to train the model. The test data will be assigned to the suffix division in the next cycle, and the train data to the last k-1, and so on. It will track the model’s performance in each iteration and then average the correctness of the findings. As a result, it’s a lengthy procedure. In our study, we will measure performance using 10-fold cross-validation, which involves training and testing the model 10 times with each set of hyper-parameter data. As a result, utilizing Grid-Search and cross-validation to find the best hyper-parameters takes a long time. If we have a large enough training set, we can get away with only utilizing a single independent validation set, but cross-validation is a better strategy to avoid over-fitting.

## 4. Results

### 4.1. Descriptive Statistics

This subsection contains two important types of statistical analysis:**Standard statistical procedures**: include the following:
Descriptive Statistics: Descriptive statistics describe the basic and essential variables of the data, such as mean, standard deviation (std), median (Med), minimum (min), 1st Quartile, or $Q_{1}$ (25%), 2nd Centile, or $Q_{2}$ (50%), 3rd Centile or $Q_{3}$ (75%), maximum (100%), and skewness and kurtosis (kur). Table 7 depicts these statistics of the various body and environmental sensor signals for all participants.Correlation Matrix: Correlation is a statistical approach for determining if and how strongly two independent variables are connected. The correlation coefficient (or “r”) is an indicator of the strength of the linear relationship between two variables and is the principal outcome of a correlation. It has a range of −1.0 to +1.0. The closer r is near +1 or −1, the closer the two variables are linked and calculated as follows:$$r=\frac{cov(x,y)}{\sigma_{x}\sigma_{y}}$$If *r* is near 0, it implies that the variables have no connection. If *r* is positive, it indicates that when one variable grows, the other gets bigger as well. If *r* is negative, it implies that while one gets bigger, the other shrinks (this is known as an “inverse” correlation). Figure 11 depicts a graphical representation of a correlation matrix.Covariance Matrix: covariance matrix on the other hand, is a square matrix that depicts the covariance between each pair of variables in a random vector. If the covariance is positive, it implies that the frequency of the two variables is increasing. However, if the correlation is negative, it indicates that the two variables are often falling. Finally, if the covariance is 0, there is no relationship between the two variables as shown in Table 8, EDA has a negative correlation with (HR, UV, bTemp) and a positive relationship with (HR, UV, b-Temp) (EnvNoise, Air pressure) and Motion has a negative relation with (bTemp) and positively related with (EDA, HR, UV, EnvNoise, air pressure) and so on according to (HR, UV, EnvNoise, Air-Pressure, bTemp).PCA: PCA (Principle component analysis) was also used to identify the relation between features included in the multiple regression analyses. The principal component analysis (PCA) is an example of a regression component analysis system (PCA). Figure 12 depicts: the first PCA component of (b-Temp, HR) on-body features exhibits positive correlations with environmental variables since they are both oriented towards the same right side of the plot (UV). On-body variables (EDA) and Motion, on the other hand, have positive coefficients with external variables (AirPressure, Env-Noise) because they are both oriented to the plot’s left-top side and have a negative relationship with b-Temp. Because (Motion, Air-Pressure) features are oriented on the top y-axis and negatively linked to bTemp, they have negative coefficients with (UV, HR, bTemp) features, whereas (EDA, EnvNoise) features have positive coefficients with (UV, HR). Consequently, we must comprehend the link between environmental and on-body factors and their influence on human emotions.**Analysis of the Poincare plot**: this is a scattergram technique that isn’t linear. A poincaré plot is a graph that shows NN(i) on the x-axis and NN(i + 1) (the next NN interval) on the y-axis, with NN intervals in between (the distance between each heartbeat). We utilized poincaré plots to display and evaluate heart rate variability (HRV) normality, excluding those with noisy heart rate patterns and assessing heart health [55].The standard deviation of the instantaneous beat-to-beat NN interval variability (minor axis of the SD1), the standard deviation of the continuous long-term RR interval variability (major-axis of SD2), and the axis ratio (SD2/SD1) based on the analysis of this plot [54]. A higher or lower heart rate variability (HRV) is determined by the ratio (SD1/SD2), with a higher proportion indicating excellent health and a lower rate indicating poor health.Given a time series, this visualisation can be calculated as follows:
$$X_{t}+X_{t+1}+X_{t+3}+.......$$Following that, return plots $\left(X_{t},X_{t+1}\right)$, then plots $\left(X_{t+1},X_{t+2}\right)$, and so on. We used Poincare plots to verify the common examples of noisy and regular HR data patterns of participants as shown in Figure 13.

After applying feature-extraction to extract features for each independent variable in our data set, feature fusion to explain three fusion levels applied on the data set and feature selection to select the most important features with which we could get higher accuracy levels for users. We spilt the data set into two parts: train and test data using ‘train-test-split’ function included in the Sklearn library. The data were trained using previous supervised ML models obtaining training model from each algorithm. After that, these algorithms are combined or ensembled using the **stacking** ensemble method. Then, this model is trained and then evaluated using new data (test-data) to obtain testing model. Finally, we test this model to obtain a predictive model with a specific accuracy.

### 4.2. Emotion Ensemble Predictive Models

In this section, we present results of Emotion predictive models that depend on combining environmental and physiological data of senors using ensemble learning. Because our features are multi-model, we applied different classifiers of ensembles techniques.

To evaluate the results of ensemble methods including these ML algorithms (SVM, KNN, DT, RF), we used cross-validation for estimating model’s performance. In subject-independent training, this processing was performed by K-fold cross-validation to evaluate the performance of the model. Cross-validation is a technique for determining how well a model generalizes to a new data set. In k-fold cross-validation, the number of folds is k. Cross-validation is used to train the model ten times (CV= 10). As a result, the k value would be 10. The classifier’s accuracy is calculated by scoring = ‘accuracy’.

Using this technique, we could get the following results mentioned in Figure 14 that depict a comparison of accuracy levels between these three ensemble methods bagging, boosting and stacking using (KNN, SVM, RF, DT) as weak learners with parameters mentioned in Section 4.4 and also using Decision Tree Classifier as a meta-model to combine predictions of weak learners with specific parameters Meta-model DT (max-depth = 200, max-leaf-nodes = 800).

Stacking ensemble method outperforms bagging and boosting methods. Such that, stacking produces an accuracy of (98.2%) with the parameters representing weak learners (KNN, RF, SVM, DT) as a first parameter and DT as a meta-classifier as a second parameter, whereas bagging and boosting methods gave (96.4%),(96.6%) accuracy levels respectively with parameters: Meta-model DT first parameter and (n-estimators = 100, random-state = 2020) as second and third parameters for Bagging method and Meta-model DT first parameter and (n-estimators = 200) as second parameter for Boosting method.

So, to create a predictive model for emotion recognition after using the feature selection approach. We decided to employ an Ensemble method called “**stacking**” to create an accurate predictive model based on this methodology (multi-learner approach) and Figure 14 [56]. We used this model to make predictions on the test set.

Stacking is distinguished from bagging in that the models are often distinct (for example, not all decision trees) and fit on the same data set (for example, instead of samples from the training data set) [57]. Rather than a series of models that correct prior models’ predictions, stacking employs a single model to learn how to best aggregate predictions from the contributing models (rather than a series of models that correct past models’ predictions) [58].

Thus, as mentioned above, to create a predictive model based on **the stacking** ensemble method we employed Support Vector Machine (SVM), Decision Tree Classifier (DT), K-Nearest Neighbors (KNN), and RandomForest Classifier (RF) as weak learners to train and model our labeled data set. We stacked the combination of these methods and used Decision Tree Classifier (DT) as the Stacking Model Learner with parameter (max-depth = 200, max-leaf-nodes = 800) to achieve better results. These algorithms have all been proved to be effective in identifying emotional reactions based on on-body and ambient sensors.

Figure 15 represents the Accuracy levels between all base learners of two modalities (Physiological with parameters: [knn(n-neighbors = 5, p = 2, weights = ‘uniform’), svm(C = 100, gamma = 0.1), DT(max-depth = 70, max-leaf-nodes = 440), RF(n-estimators = 100, max-depth = 13)] and Environmental with parameters: [knn(n-neighbors = 5, weights = ‘distance’), svm(C = 100, gamma = 0.1), DT(max-depth = 100, max-leaf-nodes = 4000), RF(n-estimators = 750, max-depth=100)]) and also accuracy levels of base classifiers based on the entire data-set with parameters: [knn(n-neighbors = 4, weights = ‘distance’), svm(C = 100, gamma = 0.01), DT(max-depth = 16, max-leaf-nodes = 800), RF(max-depth = 15, n-estimators = 100)] according to the overall stacking model of all data-set with parameters mentioned in Table 11. It’s clear that the Stacking model yielded excellent results with four classifiers and outperformed the individual classifiers of two modalities and entire data-set with an Accuracy of 98.2% (validation) and prediction accuracy (98%).

And Figure 16 and Figure 17 depict accuracy levels between base classifiers of each modality and stacking learner.

### 4.3. Performance Evaluation

In this subsection, we evaluated the training model using specific performance metrics to measure model’s Performance such as Accuracy, Precision and Recall and F1-Score.

Based on this, the results showed the improvement in the classification accuracy of the emotion prediction method by combining decision fusion and feature fusion dependent on the Stacking ensemble Learner. Table 9 displays the classification report of the Stacking Model that shows the essential classification metrics precision, recall, and F1-score and support for the five labels.

Finally, to check whether the results of the classifiers used in our research are relevant or not we used statistical tests to ensure if the results are true or not. We used t-test method to determine if there is a statistically significant difference between each modality and entire data accompanied with results of classifiers used in research.

Table 10 depicts results of classifiers for each modality and entire data-set. We also calculated a t-test paired between environmental modality and all data and a t-test paired between two modalities to test the significant difference between them. We found that the *p*-value was less than 0.05 (*p*-value < 0.05), so we rejected the null hypothesis. Thus, there is a significant difference between each modality and all data for (SVM, DT, RF, KNN) and stacking learners.

### 4.4. Hyper-Parameter Optimization

Following feature extraction to extract features for each independent variable in our dataset, feature fusion to explain three fusion levels applied to the dataset, and feature selection to select the most important features with which we could achieve higher accuracy levels. However, there were times when applying the classifiers in our study to our preprocessed dataset with those significant features resulted in over-fitting at default classifier parameters, so we decided to use **the gridsearchcv optimization method** as in Section 4.4 for each classifier in ML algorithms, by which we could reach the optimal hyper-parameters for each classifier, and N-fold cross-validation over the data of all users was performed, and then the results over each classifier were compared. This processing was performed using K-fold cross-validation to evaluate the performance of participant’s data in subject-independent training.

Four classifiers were evaluated: KNN (k-Nearest Neighbor) [59], RF (Random Forests) [60], DT (Decision Tree) [61,62], SVM (Support Vector Machine). These categorization models were created using the sklearn library, which combined standard machine learning methodologies [63]. We decided to categorize the current emotional state of all people into one of five categories: 1, happy; 2, extremely pleased; 3, neutral; 4, angry; and 5, very angry [64].

Tweaking hyper-parameters can be performed in a variety of ways. Grid search and random search are two of them. Model parameters having pre-programmed values are known as hyper-parameters [65]. For example, in a random forest, the number of trees or the penalty intensity in a Lasso regression. They’re all numbers that impact the model’s behavior and are set before the training procedure. Because we don’t know their best values in advance, we should tweak a model’s hyper-parameters. Because each hyper-parameter is different, a model with multiple hyper-parameters may perform worse or produce over-fitting or under-fitting.

Utilizing the default KNN and SVM values resulted in under-fitting, whereas using the default DT,RF values resulted in over-fitting, according to our results. Such that, KNN and SVM training and prediction results were: (72%) training, (70%) testing for KNN algorithm and (93%) training, (34%) testing for SVM algorithm. Note that these models didn’t train our data-set using default parameters enough well resulted in **under-fitting**. Whereas, DT and RF training and testing accuracy levels were: (97%) training, (100%) testing for DT algorithm and (96.7%) training, (100%) testing for RF algorithm. Based on this, a large amount of data get trained to these models resulted in **over-fitting**.

Because the model is inaccurate in both cases, we should have used this technique to determine the intermediate number of trees that leads to the best performance.

Table 11 shows the parameter information and definitions for each classifier for thirty-users in our dataset after applying Hyper-parameter optimization.

## 5. Discussion

As previously mentioned, we presented a real-world study modeling emotions fusing environmental and physiological variables. The difference between our research and previous research [12] is the experimental settings such as place, environment, culture and context. Our experiments were conducted in Minia University campus, whereas the others were conducted in Nottingham city centre and participants were shopping. Despite the differences in the working environment, it is feasible that this study has been compared to our research because the methodology (information fusion, feature fusion, decision fusion) and environmental, physiological sensor data employed are similar to us. When comparing our findings to previous research [12], we discovered that our research outperformed previous study that developed a user-dependent prediction emotion model with a prediction accuracy 98.2%, whereas previous one achieved an accuracy 86% with the same classifiers (KNN,DT,RF,SVM) as base classifiers and the stacking ensemble method and also outperformed this study [8] that used deep learning methods (CNN-LSTM) to create a predictive emotion model based on integrating on-body and ambient sensors. So, we discovered that no other study predicted this level of accuracy based on integrating environmental and on-body sensors to produce a user-independent stacking emotional predictive model of all participants employing two modalities.

As a result, this work has obtained better results for physiological and environmental components for the entire data-set and the Stacking model to build a general predictive emotion model using feature selection and stacking model withe hyper parameter optimization. In addition to building a subject-independent model for emotion detection.

The physiological modality shows higher accuracy levels among most of the base classifiers of all participants, and the environmental modality depicts lower accuracy levels, as in Figure 15. However, accuracy levels of entire data-set outperformed results of these two modalities.

As a result, the accuracy levels vary depending on the base learners [12]. In addition, we found that the stacking model of combined data of all participants exceeds the accuracy levels of single two modalities (physiological and environmental) factors and accuracy levels of the whole data-set. One of the limitations of the self-report of the user for the emotion. It can envy the data in the algorithm training.

Finally, researchers use ensemble learning when the quantity of the data to be analyzed is too large and complex for a single classifier to handle. As a result, training a classifier with a large amount of data is impractical, and finding a single technique that works effectively for all testing data is challenging [12]. In this work, the staking ensemble learning gave the best performance compared to the other classifiers for the ensemble methods.

## 6. Conclusions and Future Work

Emotion recognition is considered a vital task in HCI applications. There are many ways to express emotions. Many factors affect human emotions. Researchers used many methods to automatically detect emotions such as facial expressions, gestures, physiological reactions and many more. In this paper, we presented an information fusion approach for combining on-body and ecological sensing, which offers new possibilities for data collection and analysis. To do so, we conducted a real-world “in the wild” study that included on-body and mobile sensors. We collected data from thirty participants walking around Minia University campus.

The collected data were cleaned and aggregated, and feature fusion and feature engineering were conducted (feature extraction and feature selection). Then, predictive models were created using different combinations of body and environmental variables. Different ensemble methods such as boosting, bagging and staking of SVM, DT, KNN, RF were used to train and test the predictive model. The stacked combination of these methods using DT Classifier as the Stacking Model Learner with parameter tuning achieve the best results.

Results suggested that stacked ensemble model gave the best results for emotion recognition yielding 98.2% accuracy compared to 86% achieved in [12], the only similar data set.

Future research will add other sensor modalities such as pollution, weather conditions, and social context, to improve our understanding of the underlying links between the environment, health, and body responses. Similarly, we will look at modeling these parameters in real settings. Moreover, XAI methods can be used for providing explanations for the models. In addition, deep learning methods can also be applied for automating the feature engineering process.

## Figures and Tables

**Figure 1 sensors-22-05611-f001:**
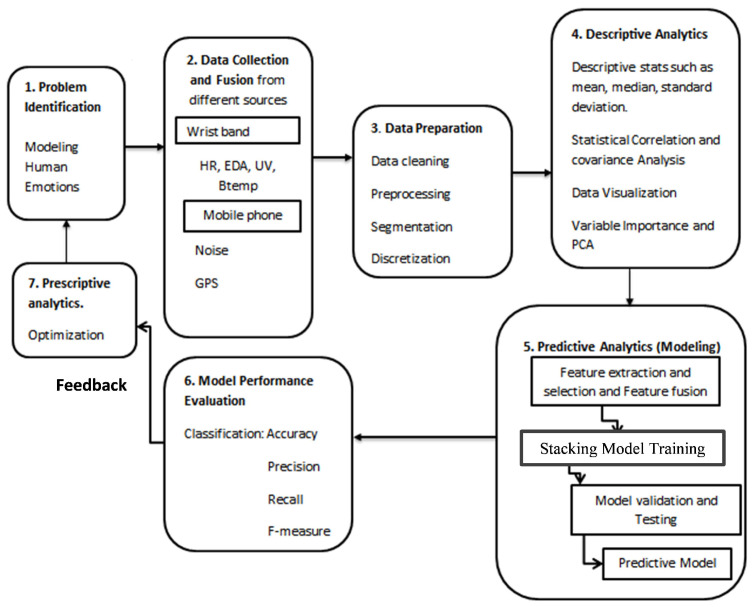
The proposed system architecture using data fusion.

**Figure 2 sensors-22-05611-f002:**
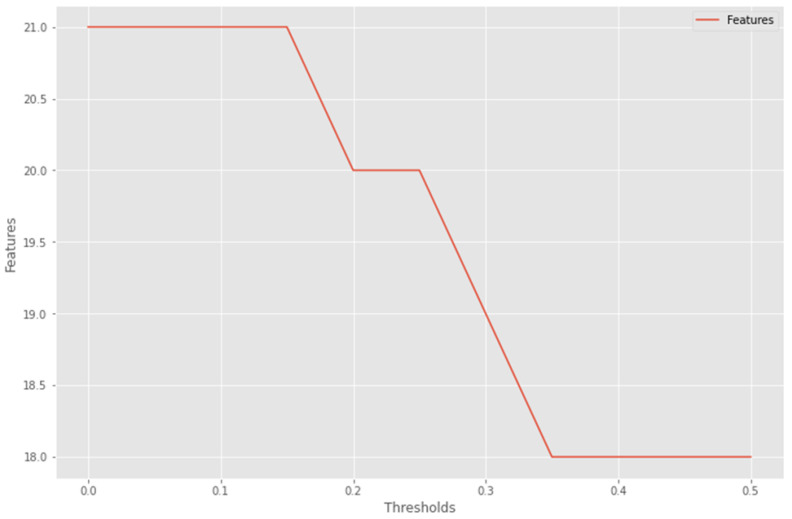
Line Plot of Variance Threshold (X) Versus Number of Selected Features to be removed (Y).

**Figure 3 sensors-22-05611-f003:**
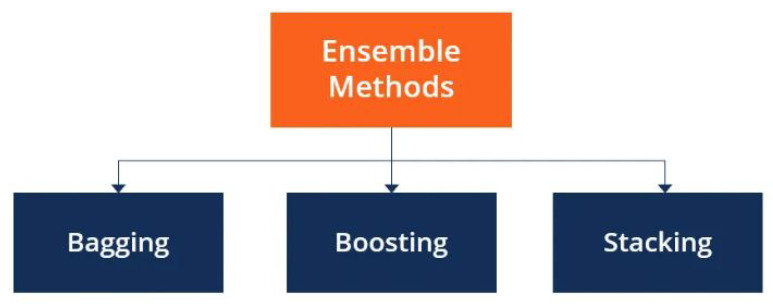
Types of ensemble methods.

**Figure 4 sensors-22-05611-f004:**
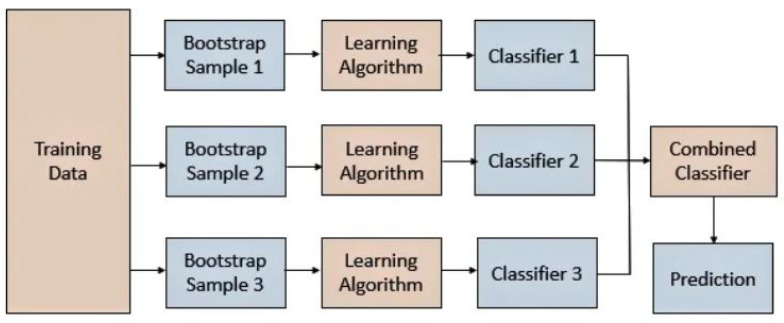
Explanation of Bootstrap Aggregating Method (Bagging).

**Figure 5 sensors-22-05611-f005:**
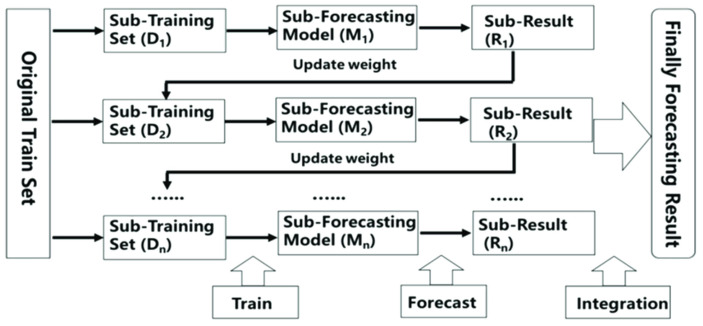
Illustration of Boosting Algorithm Architecture. “Adapted from [50]. (2021), Li, Y et al.”

**Figure 6 sensors-22-05611-f006:**
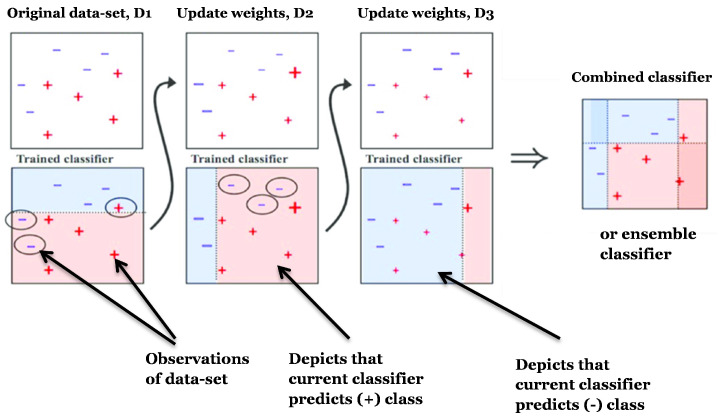
Illustration of Ada-boost Architecture Steps. “Adapted from [51]. (1997), Sodhi, A.”.

**Figure 7 sensors-22-05611-f007:**
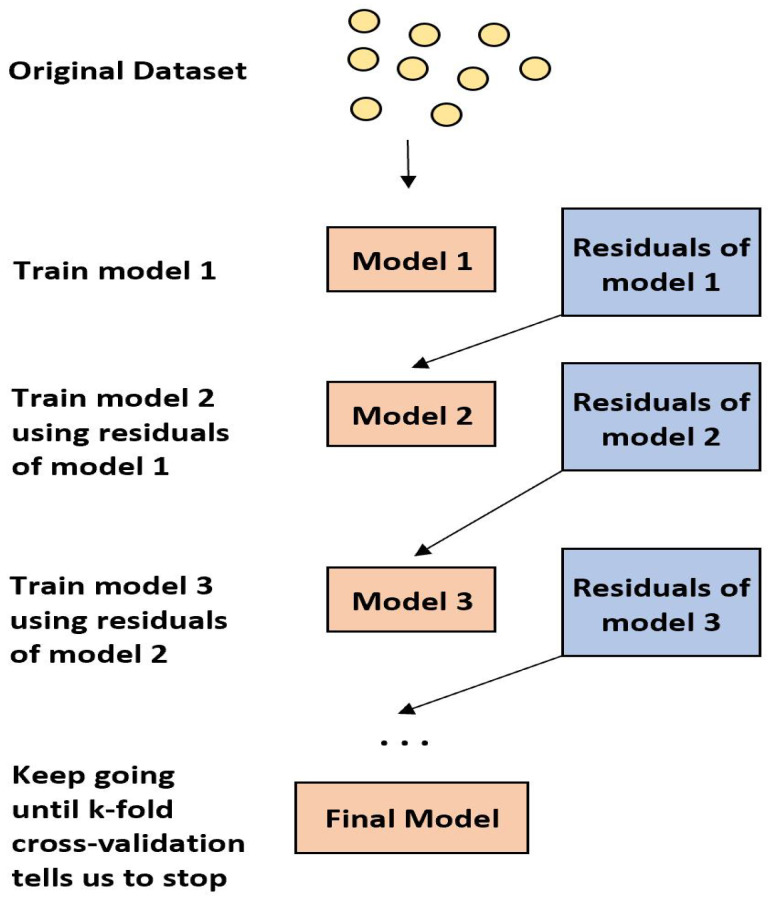
Flow Chart of Gradient boosting Algorithm.

**Figure 8 sensors-22-05611-f008:**
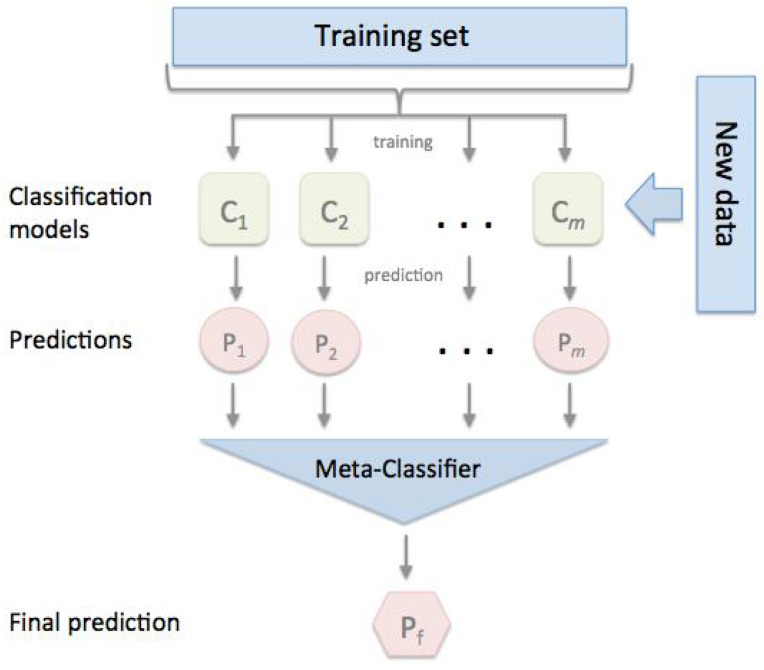
Flow Chart of Stacking Classification ensemble.

**Figure 9 sensors-22-05611-f009:**
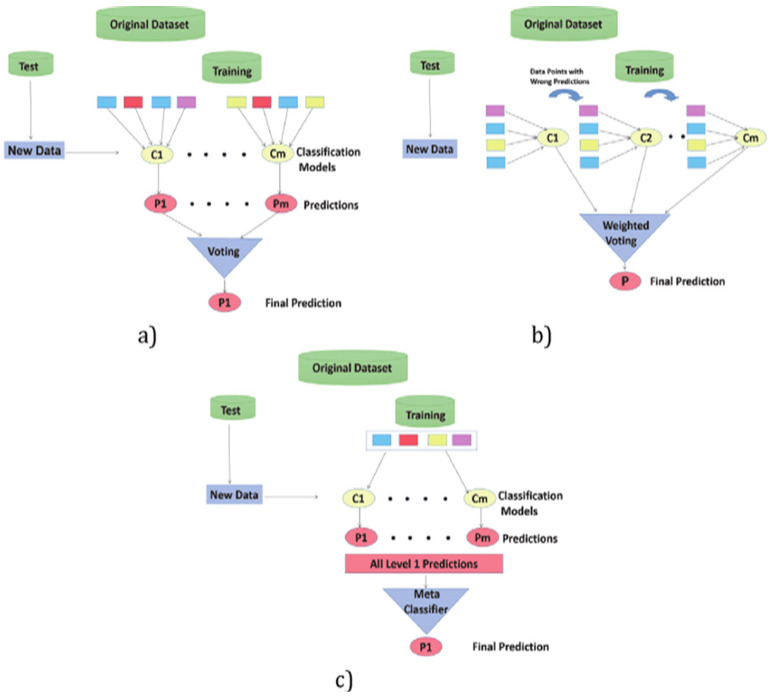
A comparison between ensemble learning methods: (**a**) Bagging Method. (**b**) Boosting Method. (**c**) Stacking Method. “Adapted from [52]. (2020), Kiyak, E.O.”

**Figure 10 sensors-22-05611-f010:**
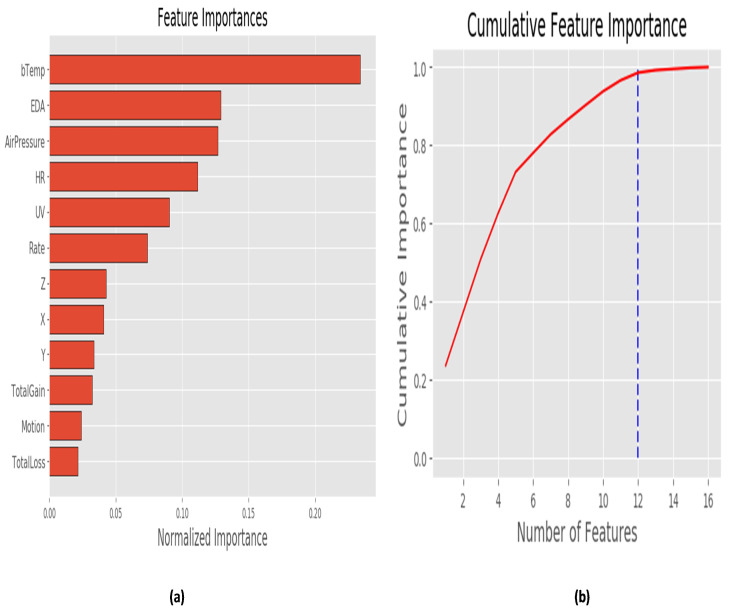
(**a**) Importance of Physiological and Environmental Features. (**b**) Cumulative Importance of These Variables.

**Figure 11 sensors-22-05611-f011:**
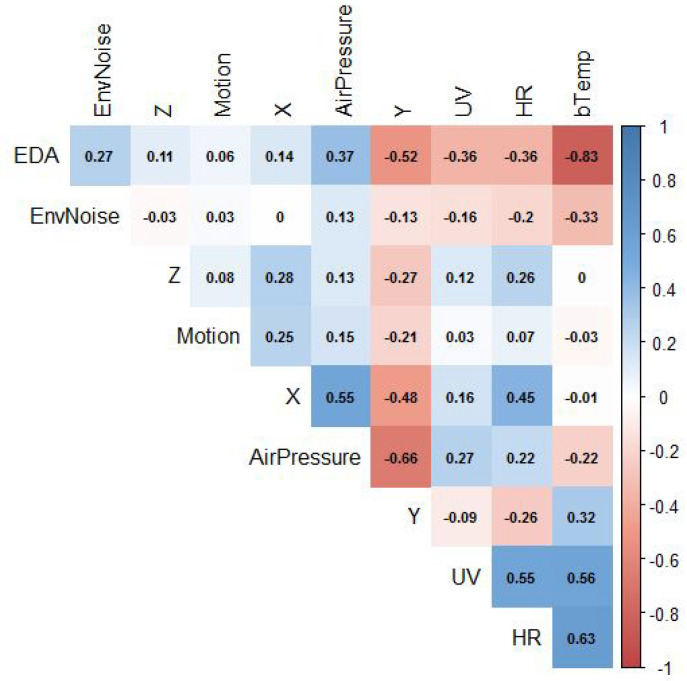
Correlation Matrix of all independent features according to dependent variable (Label).

**Figure 12 sensors-22-05611-f012:**
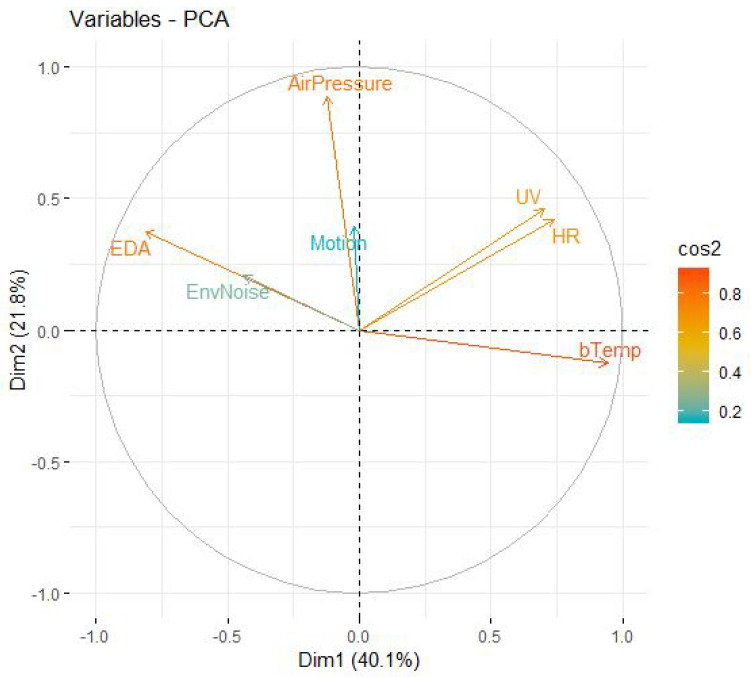
PCA plot of Body features with environmental variables.

**Figure 13 sensors-22-05611-f013:**
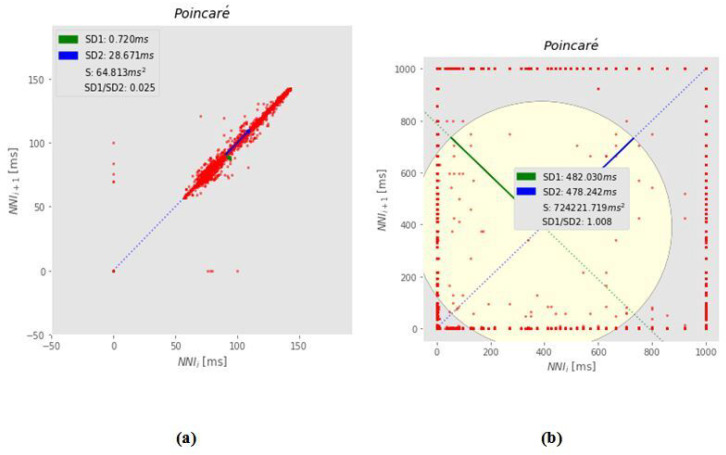
(**a**) Shows Noisy HR patterns using poincare plot. (**b**) Shows Normal HR of user after data transformation.

**Figure 14 sensors-22-05611-f014:**
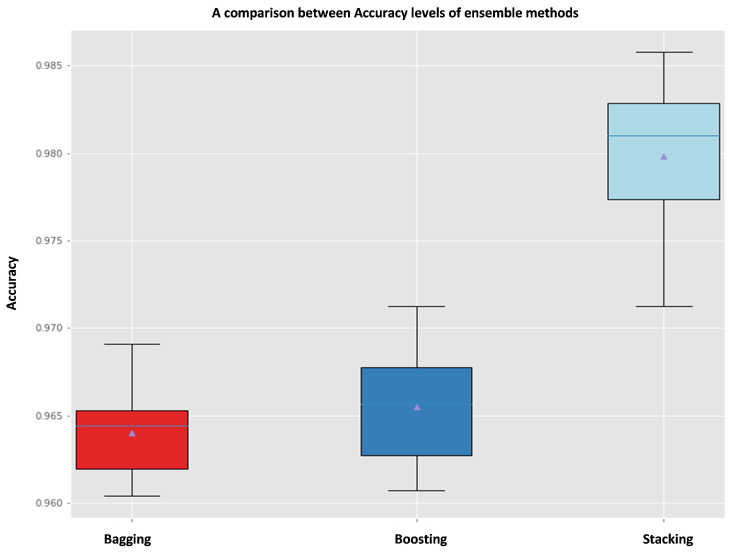
Accuracy levels comparison of Bagging, Boosting and Stacking Ensemble Methods.

**Figure 15 sensors-22-05611-f015:**
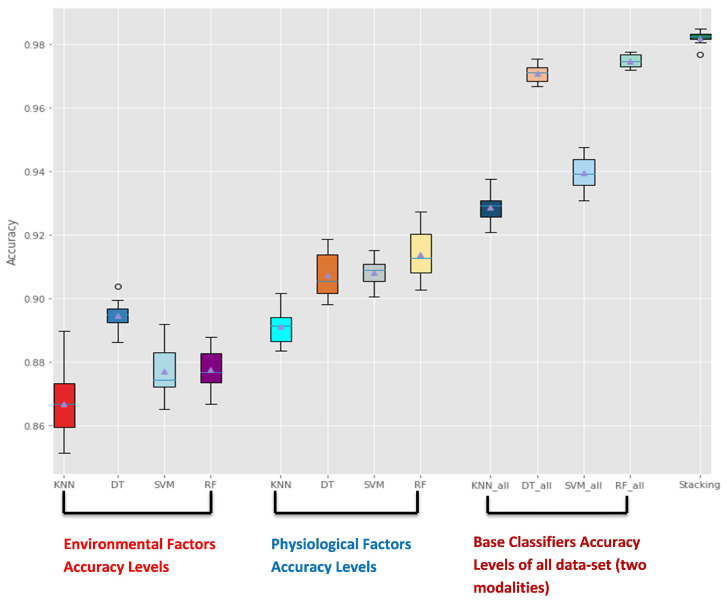
Accuracy Levels Comparison between Base classifiers Based on only Environmental, Physiological factors and those based on entire data-set with Stacking Ensemble Method Accuracy.

**Figure 16 sensors-22-05611-f016:**
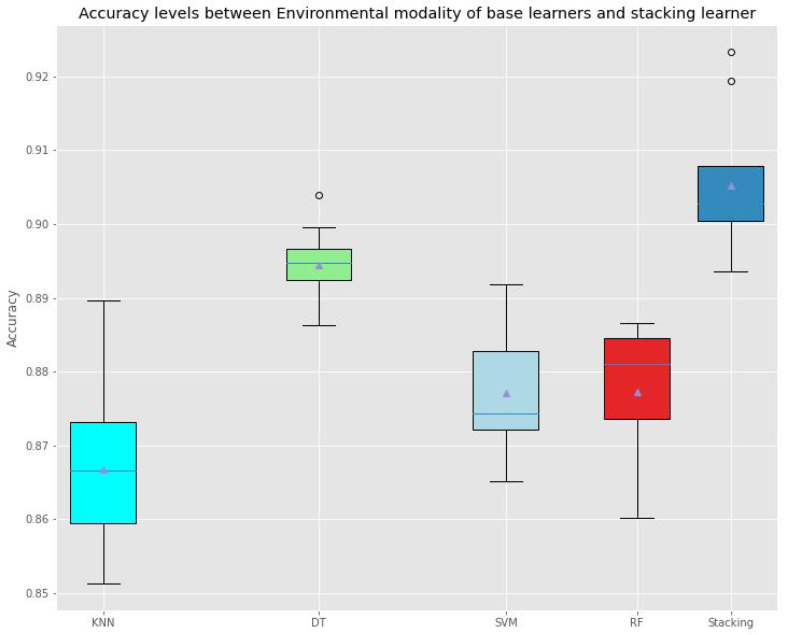
Accuracy Levels Comparison between Base classifiers of Environmental Modality and the Stacking Learner.

**Figure 17 sensors-22-05611-f017:**
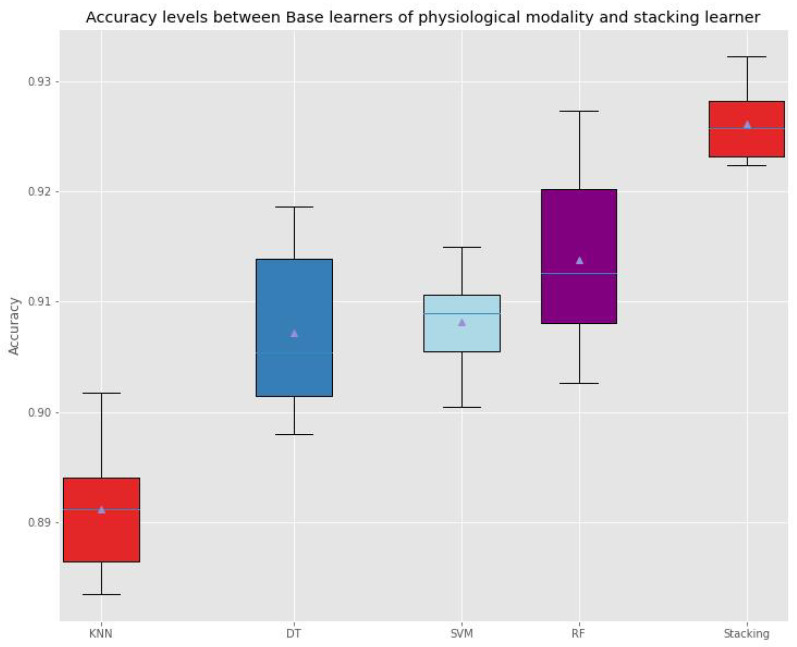
Accuracy Levels Comparison between Base classifiers of Physiological modality and the Stacking Learner.

**Table 2 sensors-22-05611-t002:** Previous Research on Recognizing Emotion from Physiological Signals and Facial Expressions.

Emotions	Measurement Methods	Data Analysis Methods	Accuracy	Ref.
Sadness, anger, stress, surprise	ECG, SKT, GSR	SVM	For recognizing three and four categories, the correct classification rates were 78.4% and 61.8%, respectively.	[33]
Sadness, anger, fear, surprise, frustration, and amusement	GSR, HRV, SKT	KNN, DFA, MBP	KNN, DFA, and MBP could classify emotions with 72.3%, 75.0%, and 84.1%, respectively	[24]
Three levels of driver stress	ECG, EOG, GSR and respiration	Fisher projection matrix and a linear discriminant	Three levels of driver stress with an accuracy of over 97%	[34]
Fear, neutral, joy	ECG, SKT, GSR, respiration	Canonical correlation analysis	The rate of correct categorization is 85.3%. Fear, neutral, and happy categorization percentages were 76%, 94%, and 84%, respectively	[35]
The emotional classes identified are high stress, low stress, disappointment, and euphoria	Facial EOG, ECG, GSR, respiration,	SVM and adaptive neuro-fuzzy inference system (ANFIS)	The total classification rates for the SVM and the ANFIS using ten fold cross-validation are 79.3% and 76.7%, respectively.	[36]
Fatigue caused by driving for extended hours	HRV	Neural network	The accuracy of the neural network is 90%	[37]
Boredom, pain, surprise	GSR, ECG, HRV, SKT	Machine learning algorithms such as linear discriminate analysis (LDA), classification and regression tree (CART), self-organizing map (SOM), and SVM	SVM produced accuracy rate of 100.0%	[38]
The arousal classes were calm, medium aroused, and activated and the valence classes were unpleasant, neutral, and pleasant	ECG, pupillary response, gaze distance	Support vector machine	The optimal classification accuracies of 68.5% for three labels of valence and 76.4% for three labels of arousal	[39]
Sadness, fear, pleasure	ECG, GSR, blood volume, pulse	Support vector regression	Recognition rate up to 89.2%	[40]
Terrible, love, hate, sentimental, lovely, happy, fun, shock, cheerful, depressing, exciting, melancholy, mellow	EEG, GSR, blood volume pressure, respiration pattern, SKT, EMG, EOG	Support Vector Machine, Multilayer Perceptron (MLP), K-Nearest Neighbor (KNN) and Meta-multiclass (MMC),	The average accuracies are 81.45%, 74.37%, 57.74% and 75.94% for SVM, MLP, KNN and MMC classifiers respectively. The best result is for ‘Depressing’ with 85.46% using SVM.	[41]
Happiness, sadness, surprise, stress	SKT, EDA, and HR	SVM, RSVM, SVM+GA, NN, DFA	The average accuracies are: 66.95% (SVM), 75.9% (RSVM), 90% (SVM+GA), 80.2% (NN), 84.7% (DFA) of this study and using Empatica E4 smartwatch to collect data from participants	[42]
Theoretical emotions	EEG signal	KNN, NB, SVM, RF, feature extraction (e.g., wavelet transform and non-linear dynamics), feature reduction (e.g., PCA, LDA)	This study achieved an average classification accuracy of over 80% and using wearable sensor to collect eeg signals	[28]

**Table 3 sensors-22-05611-t003:** Recent Previous Research on Recognizing Emotion from Physiological Signals and Facial Expressions (2021 and 2022).

Emotions	Measurement Methods	Data Analysis Methods	Accuracy	Ref.
Arousal and valence emotions. Arousal represents inactive and active emotions (Annoying, Angry, Nervous, Excited, Happy, pleased). Valence represents negative and positive emotions (Sad, Bored, Sleepy, Relaxed, Calm, Peaceful)	EEG, Facial expressions	ANN, SVM, RF, K-NN, DT, RNN, CNN, DNN, DBN, LSTM	ML classification accuracy ranges from 61.17 to 93% (SVM: 41%, ANN: 18%, RF: 14%, KNN: 9%, DT: 9%) and deep learning classification accuracy ranges from 61.25% and 97.56% (LSTM: 50%, DNN: 7%, DBN: 7%, CNN: 36%)	[43]
Arousal and valence (low and high) emotion levels.	EEG Signal	ML classifiers (KNN, SVM, LDA) and deep learning and MG3P (NN, MLP, ELM) and Gaussian process, k-means	This study performed an overall recognition rate (82.9%) [NN: 85.80%, SVM: 77.80%, KNN: 88.94%, MLP: 78.16%, 87.10%, 78.06%, 71.30%, 71.30%]	[44]
Nagtive and positive emotions	EEG signal and Facial expressions	ML classfiers: RF, KNN, SVM, DT, LDA and deep learning classifiers: CNN+LSTM	This study achieves the following accuracy levels: 63.33% RF, 63.33% SVM, 61.7% KNN, 55% DT, 51.7% LDA, 71.67% CNN+LSTM.	[45]
Negative emotions (annonyed, stressed, angry)	EGG physiological signals	ML classifiers: LR, SVM	It achieves accuracy levels: 75.00% LR, 72.62% SVM.	[46]

**Table 4 sensors-22-05611-t004:** The collected data.

Microsoft Wrist-Band 2	Android Phone 7
Heart Rate (HR)	Self-Report of Emotion (1–5)
Body-Temperature (Body-Temp )	Environmental Noise ( Env-Noise )
Electro Dermal Activities (EDA)	GPS Location (lat, lon)
Hand Acceleration (Motion as three-axis accelerometer)	
Air Pressure	
Light (UV)	

**Table 5 sensors-22-05611-t005:** Extracted and removed features.

Features Extracted	Meaning	Removed Features
EDA	It’s called Elctro-Dermal Activity, skin conductance and galvanic skin response (GSR).	FLightofStairsAscended, FLightofStairsDescended, Lat, Lng (Location)
HR	Heart Rate (Also called pulse) is the number of times the heart beats.	
Air-Pressure	The pressure of the air.	
bTemp	It’s called Body temperature.	
Env-Noise	Represents Environmental Noise.	
UV	UV means Ultra-violet radiation.	
Motion	An accelerometer with three axes. It’s a combination of (X, Y, Z) axes.	
X	Participant’s Motion in X-axis.	
Y	Participant’s Motion in Y-axis	
Z	Participant’s Motion in Z-axis	
Total-Gain	The overall gain achieved by the participant.	
Total-Loss	The amount of calories lost.	
Stepping-Gain	Steps achieved or gained during travel.	
Stepping-Loss	The steps in which a loss of calories occurred.	
Steps-Ascended	Number of steps in ascending order.	
Steps-Descended	Number of steps in descending order.	
Rate	The rate of movement in X, Y, and Z directions.	
Label	The target emotion labels can be (1–5)	

**Table 6 sensors-22-05611-t006:** A comparison between the differences and characteristics of three ensemble methods (Bagging, Boosting and Stacking).

	Bagging	Boosting	Stacking
Differences	Bagging often considers homogeneous weak learners, learns them independently from each other in parallel, and combines them following some kind of deterministic averaging process.	Boosting frequently takes into account homogeneous weak learners, trains them sequentially in a highly adaptive way (a base model depends on the preceding ones), and combines them by a deterministic method.	Stacking frequently takes into account diverse weak learners, trains them concurrently, and then combines them by training a meta-model to produce a prediction based on the output of the many weak models.
Characteristics	Bagging enables a group of weak learners to work together to outperform a single good student. Additionally, it aids in variance reduction, hence preventing the over-fitting of models during the process.	Boosting models could be improved with the help of several hyper-parameter variables. Boosting algorithms iteratively combine several weak learners and enhance observations. It might lessen a high bias that frequently appeared in models like decision trees and logistic regression. With Boosting Algorithms, characteristics are only chosen that have a large impact on the target, potentially reducing dimensionality and improving computational efficiency.	Stacking can harness the capabilities of a range of well-performing models on a classification or regression task and make predictions that have better performance than any single model in the ensemble.

**Table 7 sensors-22-05611-t007:** Descriptive statistics for the collected signals.

	Min	Q1	Med	Q2	Mean	Q3	Max	skw	kur	std
EDA	0.0	0.0	340,330	340,330	221,954	340,330	340,330	−0.6	−1.7	165,736
HR	0.0	0.0	70.0	70.0	45.7	70.0	70.0	−0.6	−1.7	34.1
UV	78.0	80.0	82.0	82.0	82.5	85.0	89.0	0.6	−1.0	3.7
X	−0.1	0.0	0.0	0.0	0.0	0.0	0.0	−0.6	−0.5	0.0
Y	0.1	0.1	0.1	0.1	0.2	0.2	0.3	0.2	−1.7	0.1
Z	0.9	0.9	1.0	1.0	1.0	1.0	1.1	−0.1	1.5	0.0
EnvNoise	49.0	52.0	52.0	52.0	52.6	53.0	56.0	0.2	−0.2	1.6
AirPressure	0.0	0.0	1010.6	1010.6	703.1	1010.7	1010.7	−0.8	−1.4	475.5
bTemp	0.0	0.0	22.8	22.8	15.8	22.8	22.8	−0.8	−1.4	10.7

**Table 8 sensors-22-05611-t008:** Covariance Matrix of all on-body and environmental variables.

	EDA	HR	UV	Motion	EnvNoise	AirPressure	bTemp
EDA	26,655,597,441	−1,012,733	−202,362,467	3430.42	95,587	33,144.6	−648,616
HR	−1,012,733	293.4	32,554.2	0.67	−7.4	2.104	51.51
UV	−202,362,467	32,554.2	11,783,951.32	40.46	−1174.20	516.51	9288.6
Motion	3430.4	0.57	40.46	0.13	0.016	0.032	−0.046
EnvNoise	95,587.03	−7.40	−1174.20	0.016	4.82	0.15	−3.40
AirPressure	33,144.58	2.10	516.51	0.032	0.15	0.31	−0.58
bTemp	−648,616	51.51	9288.58	−0.046	−3.39	−0.58	22.79

**Table 9 sensors-22-05611-t009:** Classification Report of the Stacking Model.

	Precision	Recall	F1-Score	Support
1	0.94	0.98	0.96	666
2	0.98	0.97	0.98	1661
3	0.99	0.99	0.99	2532
4	0.99	0.98	0.99	1813
5	0.98	0.99	0.99	1415
Accuracy			0.98	8087
macro avg	0.98	0.98	0.98	8087
weighted avg	0.98	0.98	0.98	8087

**Table 10 sensors-22-05611-t010:** A Comparison between results of classifiers for (each modality and entire data) and the stacking learner of them.

Classifier	Body + Environmental	Body	Environmental
KNN	93%	89%	87%
SVM	94%	91%	88%
DT	97%	91%	89.50%
RF	97.50%	91.40%	88%
Stacking	98.20%	93%	91%

**Table 11 sensors-22-05611-t011:** Parameters of Classifiers Definition used in the Stacking Model after applying Hyper-parameter Tuning.

Classifier	Parameter	Parameter Explanation
KNN	KNN parameters (weights = distance, p = 2, n-neighbors = 4, leaf-size = 30, algorithm = ’auto’)	Explanation:- **Weights**: weight function used in prediction; **p**: power parameter for the Minkowski metric; **n-neighbors**: number of neighbors to use; **leaf-size**: leaf size passed to algorithm; **algorithm**: used to compute the nearest neighbors
SVM	SVM Parameters (C = 100, gamma = 0.01)	**C**: Regularization parameter. The strength of the regularization is inversely proportional to C. Must be strictly positive. The penalty is a squared l2 penalty. C can be each value from this [1,10,100,1000]. **Gamma**: gamma is a parameter for non linear hyperplanes. The higher the gamma value it tries to exactly fit the training data set. Gamma values can be in range of [0.0001,100]. We can see that increasing gamma leads to overfitting as the classifier tries to perfectly fit the training data. **Kernel function**: Kernel Function is a method used to take data as input and transform into the required form of processing data. “Kernel” is used due to set of mathematical functions used in Support Vector Machine provides the window to manipulate the data. Kernel can be sigmoid or poly or rbf. Usually kernel function is RBF(radial-bias function).
RF	RF Parameters (n-estimators = 700, max-depth = 100)	**n-estimators**: the number of trees in the forest; **max-depth**: the maximum depth of the tree.
DT	DT Parameters (max-depth = 300, max-leaf-nodes = 900, splitter = ‘best’)	**max-depth**: the maximum depth of the tree **max-leaf-nodes**: maximum number of leaf nodes Grow a tree with max-leaf-nodes in best-first fashion. Best nodes are defined as relative reduction in impurity. If None then unlimited number of leaf nodes. **splitter**: the strategy used to choose the split at each node.

## Data Availability

Data is available up on a reasonable request from the corresponding author.

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
