# Peer review of "Evaluating Ensemble Learning Methods for Multi-Modal Emotion Recognition Using Sensor Data Fusion"

_sensors, 2022, doi:10.3390/s22155611_

Round 1
Reviewer 1 Report
The article presents an emotion recognition system based on machine learning and the acquisition of several sensors, monitoring environmental parameters and signals obtained from the body itself. The idea is interesting, but it needs a maturing proposal and a considerable improvement in writing and adaptation to the journal's template. Some points need to be considered.
Moreover, it is not a data fusion feature because the signal acquired were not combined to merge on a new feature or in a new parameter. The authors performed the data fusion on acquisition because the ensemble is a classifier model by itself.
Major:
- The Ethical Committee number is not present. Does an Ethical Committee pass the research? Because lines 190-193 mention the subjects' inscription but do not mention their approval of the research.
- How many days were the data collected from the volunteers?
- It needs a Figure with the sensors placed on the volunteers aiming to replicate this work.
- How do the researchers guarantee the labels of the volunteers if the data was not verified online continually?
- The authors are not clear about the emotions they want to classify. This information appears online 426. Why a state of “no-emotion” was not considered?
- The figures are of low quality. Some charts do not have a legend on their axes, and the legends are incomplete. In addition, some figures are classical on texts that explained the algorithms, as in Figures 5 and 8. The authors should create their figures and not use other figures or ideas without their owner's permission.
- Figure 2: Before showing the number of features that will be removed, it is necessary to show which features were extracted and removed.
- Explain the terms of equation (1). Why only X the root is extracted?
- The author did not explain the actual size of the database explicitly.
- Why was a comparison with recent deep learning models not conducted because these classifiers presented high accuracy in the pattern recognition process?
- Section 4 is confusing and should be revised entirely.
- Did the hyper-parameter optimization allow for the results section?
- Lines 436-437 – “(…) resulted in over-fitting, according to our results”. It should be demonstrated.
- The authors comment in the discussion that there are few related works because they decided to evaluate environmental parameters. However, several works evaluate individual sensors for this exact application. These works must also be inserted and discussed to strengthen the results obtained.
- An evaluation was not performed if the ensembles were evaluated with the classifiers separately.
- Why was not a feature selection method with clear methodologies, such as SFS, used? What is the gain obtained between the feature selection methods?
- It is missing a statistical analysis to verify if the results obtained from the classifiers are relevant.
Minor:
- The abstract should be rewritten using the instructions presented on the journal’s template.
- Review the sentences:
- “Collecting the dataset is coming (…).”;
- There are phrases without punctuation (as in 189);
- Lines 208 – 218: the text is misspelled and poorly formatted;
- Line 227, begging of the phrase;
- Some sentences need references to the embassy the affirmations, as the first paragraphs of the introduction.
- Remove the spaces on the parenthesis, as in line 81 for (HR) and (EDA).
- Line 105: in the text, the name of the authors are the wrong place. Only one author is mentioned, and if is a need to mention the paper’s authors, it should be done correctly.
- Lines 121 and 122 have some phrases that did not connect with the paragraphs. In addition, lines 136 to 150 are disconnected from the text and are out of the template. Please, revise them.
- What is the RR in table 1? Moreover, in Table 1, correct “(…) studies, They use (…)”, “sometimes called (…)”, “Galvanic Skin Resistance (GSR)”, revise the equation on motion, because if this equation is the root mean square, all the variables should be inserted on the root square.
- On the text, as in Table 1, rewrite all the sentences in which there are “They” to mention some work or author. Remove the pronouns and correct the sentences.
- The authors commented on the difference between bagging, boosting, and stacking, but did not separate their differences and characteristics.
- Appendixes I and J should be inserted in the text because they have relevant information about the system.
- Appendix H is useless, and its information can be merged into the text.
- Section 3.5.1 are about ensemble learning methods, but this text is not related to this section.
Author Response
Response to Reviewer 1 (Major) Comments
Dear Reviewer,
First, thank you for your time, valuable comments, and contributions to improve the manuscript. Following are the comments and their responses.
Point 1: The Ethical Committee number is not present. Does an Ethical Committee pass the research? Because lines 190-193 mention the subjects' inscription but do not mention their approval of the research.
Response 1:
Yes, we did an Ethical Committee Approval at Minia University [MU-FCI-22-1]. We thought it is not necessary to put it in paper. But, we now added it.
Point 2: How many days were the data collected from the volunteers?
Response 2:
The data was collected across around 15 days, as we were satisfied with only two participants per day so that each participant could collect a huge amount of data in fresh weather (starting at 10 am) while wandering around the university campus during a period not exceeding an hour.
Point 3: It needs a Figure with the sensors placed on the volunteers aiming to replicate this work.
Response 3: This figure depicts a snapshot of sensors extracted from wristband 2 which is wirelessly connected to smartphone app software known as 'EnvBodySens'. It is already existing in our previous research.
Kanjo, Eiman, Eman MG Younis, and Nasser Sherkat. "Towards unravelling the relationship between on-body, environmental and emotion data using sensor information fusion approach."
Information Fusion 40 (2018): 18-31.
Point 4: How do the researchers guarantee the labels of the volunteers if the data was not verified online continually?
Response 4: If we understood correctly, our data-set is saved automatically including 5-emotion labels ranging from positive to negative using smartphone app software known as 'EnvBodySens' which is wirelessly connected to the Microsoft wristband 2 (wearable sensor) with a time stamp and date. The labels were verified offline after data collection.
Point 5: The authors are unclear about the emotions they want to classify. This information appears online 426. Why a state of “no emotion” was not considered?
Response 5: Because our data-set was collected using a wearable sensor called ‘Microsoft wrist-band 2’ which is wirelessly connected to smartphone app software called ‘'EnvBodySens' which has an interface consisting of five labels of emotions [1-5] [1-happy or positive, 2- very happy or very positive, 3- neutral, 4- negative or angry, 5- very negative of very angry] (represented by buttons on the app interface) that enable the participant to choose only one of the five emotion classes. So, a state of ‘no emotion’ doesn’t exist from the options of the app.
Point 6: The figures are of low quality. Some charts do not have a legend on their axes, and the legends are incomplete. In addition, some figures are classical on texts that explained the algorithms, as in Figures 5 and 8. The authors should create their figures and not use other figures or ideas without their owner's permission.
Response 6: we checked it and updated the quality of figures and tried to change the caption on figures and add legends to some figures as in Fig 2 and Fig 13 in our research. And, we added some references to some figures used in ensemble methods.
Point 7: Figure 2: Before showing the number of features that will be removed, it is necessary to show which features were extracted and removed.
Response 7: At figure 2, we depicted the number of features to be removed based on data pre-processing steps. Based on this, the following table depicts the features that were extracted and removed. We added them after fig 2 in table 4..
|
Features extracted |
Features removed |
|
|
EDA, HR, UV, EnvNoise, Air-pressure, bTemp, X, Y, Z, SteppingGain, SteppingLoss, StepsAscended, StepsDescended, TotalGain, TotalLoss, Rate, Label. |
|
Point 8: Explain the terms of equation (1). Why only X the root is extracted?
Response 8: Here, motion is a three-axis accelerometer that depicts the motion of participants in the x-axis, y-axis, and z-axis because modern accelerometers incorporate tri-axial micro-electro-mechanical systems (MEMS) to record three-dimensional acceleration, so the motion equation is as follows:-
Motion= X2+Y2+Z2
Motion is the root mean square of all three components. The root is on all three components not only on X. We checked it in the paper and updated it.
Point 9: The author did not explain the actual size of the database explicitly.
Response 9: our data set consists of 30-files of thirty participants such that each file consists of 22 features including on-body sensors and environmental factors and other variables with a total size of approximately (82,729) samples.
Point 10: Why was a comparison with recent deep learning models not conducted because these classifiers presented high accuracy in the pattern recognition process?
Response 10: We checked this point in our research. We compared our results with deep learning methods used in emotion recognition based on multi-model fusion in this paper
Kanjo, E., Younis, E. M., & Ang, C. S. (2019). Deep learning analysis of mobile physiological, environmental, and location sensor data for emotion detection. Information Fusion, 49, 46-56.
Point 11: Section 4 is confusing and should be revised entirely.
Response 11: It has been rearranged.
Point 12: Did the hyper-parameter optimization allow for the results section?
Response 12: It has been added as a subsection of the results section.
Point 13: Lines 436-437 – “(…) resulted in over-fitting, according to our results”. It should be demonstrated.
Response 13: Briefly, in our research, we used four machine learning algorithms such as support vector machine (SVM), K-nearest Neighbor (KNN), Random Forest (RF), and Decision tree (DT) in addition to ensemble methods to create a user-dependent emotion model based on integrating on-body and environmental factors. When training SVM, and KNN with their default parameters for example on our data-set and then testing these algorithms on new data, we found that these two algorithms cause as we said ‘underfitting’ because training results are higher than testing results which explains that these algorithms learned well on our data-set but they couldn’t predict labels well. Whereas, when training RF, and DT on our data-set with also their default parameters and then testing on new data, we found that these algorithms cause ‘overfitting’ because testing results are higher than training results which means that these algorithms were trained too much on our dataset. Look at section 5. So, we decided to use GridSearch hyper-parameter optimization method to optimize or estimate the parameters of each classifier to avoid underfitting or overfitting. For example:
- For KNN, we tried to update parameters (n_neighbors and weights)
- For SVM, update parameters (C and gamma)
- For DT, update parameters (max_depth, max_leaf_nodes)
- For RF, update parameters (n_estimators, max_depth)
Point 14: The authors comment in the discussion that there are few related works because they decided to evaluate environmental parameters. However, several works evaluate individual sensors for this exact application. These works must also be inserted and discussed to strengthen the results obtained.
Response 14: We checked this point in our study, look at section 2 and table 2 include previous works that explain using individual sensors such as physiological signals and facial expressions for emotion recognition. But, the scarcity of similar work and data was limited in comparing the results.
Point 15: An evaluation was not performed if the ensembles were evaluated with the classifiers separately.
Response 15: we added it to the paper. We applied the stacking ensemble method to each modality and did a comparison between the base classifiers of each modality and the stacking learner. It is available in the results section.
Point 16: Why was not a feature selection method with clear methodologies, such as SFS, used? What is the gain obtained between the feature selection methods?
Response 16: We used the ‘SelectKBest’ Method for feature selection in our research because it was the best method for us such that it help us to choose the number of features at which we got a higher accuracy. This method is easy to use and has a few steps:-
- First, Scikit-learn API includes a function called feature selection that provides SelectKBest class for extracting the best features of a given dataset as follows:
from sklearn.feature_selection import SelectKBest
- Second, we called a class called ‘f-classif’ as a scoring function from the feature_selection function included in sklearn as
from sklearn.feature_selection import f_classif
- Then, we begin to select the target number of features that
are defined by the k parameter using the scoring function ‘f_classif’
select = SelectKBest(score_func=f_classif, k=n_features)
- Finally, we will fit and transform method on training x and y data
select.fit(X_train, Y_train)
X_train_poly_selected1 = select.transform(X_train)
and for test
select.fit(X_test, Y_test)
X_test_poly_selected1 = select. transform(X_test)
we didn’t add these steps in our paper because we thought that the steps aren’t necessary because we depicted which features were selected from this method.
Feature selection methods are very useful such that these methods play a vital role in creating an effective predictive model. Because they enable ML algorithms to train faster and improve the accuracy of a model because they can choose the most significant features that help us get higher accuracy.
Point 17: It is missing a statistical analysis to verify if the results obtained from the classifiers are relevant.
Response 17: we used t-test statistical to check if the results obtained from the classifiers are significantly different. we did a t-test paired between the results of classifiers. We found that there is a statistical significance between the results of classifiers with a P-value <0.05.
(Minor) Comments
Point 1: The abstract should be rewritten using the instructions presented on the journal’s template.
Response 1: We checked it in our paper and updated it.
Point 2: Review the sentences:
- “Collecting the dataset is coming (…).”;
- There are phrases without punctuation (as in 189);
- Lines 208 – 218: the text is misspelled and poorly formatted;
- Line 227, begging of the phrase;
Response 2: We checked them in the paper and updated them.
Point 3: Some sentences need references to the embassy the affirmations, as the first paragraphs of the introduction.
Response 3: We checked them and added some references to these sentecnes.
Point 4: Remove the spaces on the parenthesis, as in line 81 for (HR) and (EDA).
Response 4: we checked it and remove spaces on the parenthesis for HR, EDA
Point 5: Line 105: in the text, the name of the authors are the wrong place. Only one author is mentioned, and if is a need to mention the paper’s authors, it should be done correctly.
Response 5: we checked it in our paper and Corrected it.
Point 6: Lines 121 and 122 have some phrases that did not connect with the paragraphs. In addition, lines 136 to 150 are disconnected from the text and are out of the template. Please, revise them.
Response 6: we revised them in our paper and updated them.
Point 7: What is the RR in table 1? Moreover, in Table 1, correct “(…) studies, They use (…)”, “sometimes called (…)”, “Galvanic Skin Resistance (GSR)”, revise the equation on motion, because if this equation is the root mean square, all the variables should be inserted on the root square.
Response 7: According to RR, the RR interval is the time elapsed between two successive R waves of the QRS signal on the electrocardiogram and measured in milliseconds (ms). Please, look at this figure to identify RR-interval.
According to the updates, we checked it and updated it. We also updated the equation of motion as follows:-
Motion= X2+Y2+Z2
Point 8: On the text, as in Table 1, rewrite all the sentences in which there are “They” to mention some work or author. Remove the pronouns and correct the sentences.
Response 8: we removed the pronouns and corrected the sentences.
Point 9: The authors commented on the difference between bagging, boosting, and stacking, but did not separate their differences and characteristics.
Response 9: We checked it on paper. Table 6 depicts the differences and characteristics of these methods
Point 10: Appendixes I and J should be inserted in the text because they have relevant information about the system.
Response 10: We checked this point in our research, we inserted them in system architecture and descriptive statistics subsections.
Point 11: Appendix H is useless, and its information can be merged into the text.
Response 11: we merged the information in this appendix into the text.
Point 12: Section 3.5.1 are about ensemble learning methods, but this text is not related to this section.
Response 12: we checked it in our paper and Corrected it.
Reviewer 2 Report
The manuscript proposed the multi-model emotion classification based on ensemble learning methods and data fusion from Microsoft wristband two and Android phone 7. The collected data are heart rate, self-report of emotion, body temperature, environmental noise, electrodermal, activities, GPS location, hand acceleration, air pressure, and light. From some algorithms (k nearest neighbor, decision tree, random forest, support vector machine), the result showed good accuracy of 98.2%. This manuscript can be published on Sensors after minor revision:
- Authors should refer to some studies in 2 past years. CNN has attracted many researchers and demonstrated this is a good algorithm for emotion classification; why did the authors choose the random forest, SVM, etc.?
- Please list 22 features and 18 features of data.
Author Response
Response to Reviewer 2
Dear Reviewer,
First, thank you for your time, valuable comments, and contributions to improve the manuscript. Following are the comments and their responses.
Point 1: Authors should refer to some studies in 2 past years. CNN has attracted many researchers and demonstrated this is a good algorithm for emotion classification; why did the authors choose the random forest, SVM, etc.?
Response 1: we added some studies about physiological signals and facial expressions in 2020,2021,2022 in table 2 because we didn’t find recent studies discussing a real-world study based on data fusion of environmental and physiological sensors and their impacts on emotion labels except for two studies in 2018 and 2019 as mentioned in section 2.
We used random forest, support vector machine, k-nearest neighbor, and decision trees in our research as base classifiers to create a user-independent emotion model because they achieved higher accuracy than deep learning methods. They achieved the following accuracy levels: 97.5\% (RF), 97\% (DT), 94\% (SVM), 93\% (KNN). And when applying the stacking method on these classifiers, it achieved an accuracy of 98.2\%. This accuracy is so high that we don’t have to use deep learning methods to improve our results. Although deep learning achieved good performance so far and it omits feature engineering, they suffer from the interpretability of the model.
Point 2:: Please list 22 features and 18 features of data.
Response 2: they are listed in this table and table in the paper
|
Total Features (22 features) |
Features after preprocessing (18 features) |
|
EDA, HR, UV, Motion,EnvNoise, AirPressure bTemp, X, Y, Z, SteppingGain, SteppingLoss, StepsAscended, StepsDescended, TotalGain , TotalLoss, Rate, FlightofStairsAscended, FlightofStairsDescended, Lat, Lng, Label |
EDA, HR, UV, Motion,EnvNoise, AirPressure bTemp, X, Y, Z, SteppingGain, SteppingLoss, StepsAscended, StepsDescended, TotalGain , TotalLoss, Rate,Label |
Round 2
Reviewer 1 Report
The quality of paper increased. The authors highlighted their contribution on introduction and in the related works section. The results section are expanded with more detailed, specially about the proposed algorithms. I have some points for correction.
- In some enumerations, commas are missing (e.g., lines 6, 11, 18 and 36)
- Please, correct “We” on lines 65 , 158, 215.
- Line 68: Could not the self-report of the user for the emotion envied the data in the algorithm training? It could be more detailed on discussion.
- Line 94: insert the year on “Park and Farr” reference.
- Lines 105-107 defined data fusion but the authors did not insert a reference. Please, correct it.
- There are some words on the text on bold. Please, correct them.
- Lines 134 and 289: insert a space between the words.
- Line 146: Table’s reference is missing.
- Line 150: Table 2.
- Line 165: [Feature Level].
- Line 184: Table 3.
- Line 207: Table 4;
- Explain more the features extracted on Table 4.
- Some figures have edges (e.g, figures 5 and 6)
- Correct line 272.
- Revise the sentence in 292-296.
- Line 349 – Table 6.
- The equation on line 356 could be out of the line.
- Figure 14 – The axes titles are not the same font number of the text.
- Line 452 – “To check”;
- Line 456 – Tables are nor referenced.
- Figure 10 had the axes title comprised. Please, redraw the graphics.
- Insert the references on Table 10.
- Correct the capitulars on line 517.
Author Response
Response to Comments
Dear Reviewer,
First, thank you for your time, valuable comments, and contributions to improving the manuscript. Following are the comments and their responses. All corrections are colored in blue in the paper.
Point 1: In some enumerations, commas are missing (e.g., lines 6, 11, 18 and 36)
Response 1: We corrected them and added missing commas in these lines.
Point 2: Please, correct “We” on lines 65 , 158, 215.
Response 2: We checked them and corrected these lines.
Point 3: Line 68: Could not the self-report of the user for the emotion envied the data in the algorithm training? It could be more detailed on discussion.
Response 3: Added to the discussion.
Point 4: Line 94: insert the year on “Park and Farr” reference.
Response 4: We inserted the year on Park and Farr ref.
Point 5: Lines 105-107 defined data fusion but the authors did not insert a reference. Please, correct it.
Response 5: We inserted references in lines 105-107 in our paper.
Point 6: There are some words on the text on bold. Please, correct them.
Response 6: We checked and corrected them in our paper.
Point 7: Lines 134 and 289: insert a space between the words.
Response 7: We updated these lines and inserted a space between the words.
Point 8: Line 146: Table’s reference is missing.
Response 8: We checked the table’s reference on line 146.
Point 9: Line 150: Table 2.
Response 9: We checked and correc it in our paper.
Point 10: Line 165: [Feature Level].
Response 10: We updated it in our paper.
Point 11: Line 184: Table 3.
Response 11: We updated it in our paper.
Point 12: Line 207: Table 4;
Response 12: We checked Table 4 in our paper at line 207.
Point 13: Explain more the features extracted on Table 4.
Response 13: We explained these features in the table. Table corrected.
Point 14: Some figures have edges (e.g, figures 5 and 6)
Response 14: We checked all figres and removed existing edges.
Point 15: Correct line 272.
Response 15: We corrected this line in our paper.
Point 16: Revise the sentence in 292-296.
Response 16: We revised these sentences in our paper.
Point 17: Line 349 – Table 6.
Response 17: We checked it in our paper.
Point 18: The equation on line 356 could be out of the line.
Response 18: We checked it in our paper.
Point 19: Figure 14 – The axes titles are not the same font number of the text.
Response 19: We checked and corrected it in our paper.
Point 20: Line 452 – “To check”;
Response 20: We checked it in our paper and updated this sentence.
Point 21: Line 456 – Tables are nor referenced.
Response 21: We checked them in our paper.
Point 22: Figure 10 had the axes title comprised. Please, redraw the graphics.
Response 22: We redrew this figure in our paper.
Point 23: Insert the references on Table 10.
Response 23: We updated them in our paper.
Point 24: Correct the capitulars on line 517.
Response 24: We updated them on line 517 and also lines 518-523 in our paper.